# A Novel Bispecific T-Cell Engager (CD1a x CD3ε) BTCE Is Effective against Cortical-Derived T Cell Acute Lymphoblastic Leukemia (T-ALL) Cells

**DOI:** 10.3390/cancers14122886

**Published:** 2022-06-11

**Authors:** Caterina Riillo, Daniele Caracciolo, Katia Grillone, Nicoletta Polerà, Franca Maria Tuccillo, Patrizia Bonelli, Giada Juli, Serena Ascrizzi, Francesca Scionti, Mariamena Arbitrio, Mariangela Lopreiato, Maria Anna Siciliano, Simona Sestito, Gabriella Talarico, Eulalia Galea, Maria Concetta Galati, Licia Pensabene, Giovanni Loprete, Marco Rossi, Andrea Ballerini, Massimo Gentile, Domenico Britti, Maria Teresa Di Martino, Pierosandro Tagliaferri, Pierfrancesco Tassone

**Affiliations:** 1Department of Experimental and Clinical Medicine, Magna Græcia University, 88100 Catanzaro, Italy; caterina.riillo1@studenti.unicz.it (C.R.); d.caracciolo@unicz.it (D.C.); k.grillone@unicz.it (K.G.); nicoletta.polera@studenti.unicz.it (N.P.); giadajuli@libero.it (G.J.); ascrizzi.serena@studenti.unicz.it (S.A.); mariangela.lopreiato@gmail.com (M.L.); mariaanna.siciliano1@studenti.unicz.it (M.A.S.); rossim@unicz.it (M.R.); teresadm@unicz.it (M.T.D.M.); tagliaferri@unicz.it (P.T.); 2Istituto Nazionale Tumori IRCCS-Fondazione G. Pascale, 80131 Napoli, Italy; f.tuccillo@istitutotumori.na.it (F.M.T.); p.bonelli@istitutotumori.na.it (P.B.); 3Institute of Research and Biomedical Innovation (IRIB), Italian National Council (CNR), 98164 Messina, Italy; scionti@unicz.it; 4Institute of Research and Biomedical Innovation (IRIB), Italian National Council (CNR), 88100 Catanzaro, Italy; mariamena.arbitrio@irib.cnr.it; 5Department of Medical and Surgical Sciences, Magna Græcia University, 88100 Catanzaro, Italy; sestitosimona@unicz.it (S.S.); pensabene@unicz.it (L.P.); 6Immunotransfusion Service Unit, Pugliese-Ciaccio Hospital, 88100 Catanzaro, Italy; tigani@libero.it; 7Pediatric Hemato-Oncology Unit, Pugliese-Ciaccio Hospital, 88100 Catanzaro, Italy; eulgal@yahoo.it (E.G.); mcgalati@aocz.it (M.C.G.); 8Department of Health Sciences, Magna Graecia University, 88100 Catanzaro, Italy; loprete@unicz.it (G.L.); britti@unicz.it (D.B.); 9BiovelocITA srl, 20122 Milan, Italy; aballerinimd@gmail.com; 10Hematology Unit, Annunziata Hospital, 87100 Cosenza, Italy; massim.gentile@tiscali.it; 11College of Science and Technology, Temple University, Philadelphia, PA 19122, USA

**Keywords:** T-ALL, acute lymphoblastic leukemia, ALL, CD1a, bispecific T cell engager, BTCE, BiTE, immunotherapy, hematological malignancies, translational medical research

## Abstract

**Simple Summary:**

T-cell acute lymphoblastic leukemia (T-ALL) is an aggressive and still orphan hematologic malignancy. No effective immunotherapeutic strategies are presently available for this poor prognosis disease. We here report the development and the preclinical evaluation of a novel bispecific T-cell engager (BTCE) that simultaneously targets CD1a and CD3ε (CD1a x CD3ε), therefore recruiting T cells against T-ALL cells. We demonstrate that this CD1a x CD3ε BTCE induces activation, proliferation, and cytokine release by T cells in co-cultures with CD1a expressing T-ALL cells, resulting in a concentration-dependent killing of leukemic cells in vitro. Moreover, CD1a x CD3ε BTCE inhibits the in vivo growth of human T-ALL xenografts and improves survival of immunocompromised mice reconstituted with human PBMC from healthy donors. We believe that this BTCE is suitable for clinical development for the treatment of CD1a-expressing T-ALL patients.

**Abstract:**

T-cell acute lymphoblastic leukemia (T-ALL) is an aggressive malignancy burdened by poor prognosis. While huge progress of immunotherapy has recently improved the outcome of B-cell malignancies, the lack of tumor-restricted T-cell antigens still hampers its progress in T-ALL. Therefore, innovative immunotherapeutic agents are eagerly awaited. To this end, we generated a novel asymmetric (2 + 1) bispecific T-cell engager (BTCE) targeting CD1a and CD3ε (CD1a x CD3ε) starting from the development of a novel mAb named UMG2. UMG2 mAb reacts against CD1a, a glycoprotein highly expressed by cortical T-ALL cells. Importantly, no UMG2 binding was found on normal T-cells. CD1a x CD3ε induced high T-cell mediated cytotoxicity against CD1a+ T-ALL cells in vitro, as demonstrated by the concentration-dependent increase of T-cell proliferation, degranulation, induction of cell surface activation markers, and secretion of pro-inflammatory cytokines. Most importantly, in a PBMC-reconstituted NGS mouse model bearing human T-ALL, CD1a x CD3ε significantly inhibited the growth of human T-ALL xenografts, translating into a significant survival advantage of treated animals. In conclusion, CD1a x CD3ε is a novel BTCE highly active against CD1a-expressing cortical-derived T-ALL cells suitable for clinical development as an effective therapeutic option for this rare and aggressive disease.

## 1. Introduction

T-cell acute lymphoblastic leukemia (T-ALL) is a hematologic malignancy characterized by abnormal proliferation of T-cell progenitors, which leads to progressive infiltration of bone marrow and lymphoid organs and spread of immature leukemic T-cells in the peripheral blood [1,2].

Although intensive chemotherapy regimens have improved the prognosis of responding patients, at least 20% of pediatric and 50% of adult relapsed/refractory cases have a poor outcome [3,4]. In this setting, nelarabine is still the only new drug approved for relapsed/refractory patients with a complete response rate of 31% and 1-year overall survival rate of only 37% [4]. For all eligible patients, allogeneic hematopoietic cell transplantation (HCT) with graft-versus-leukemia remains the best curative option, demonstrating the efficacy of harnessing the immune system against leukemic cells [5,6]. Unfortunately, in contrast to the huge and recent advancement in the treatment of relapsed/refractory B-ALL patients, where specific targeting of B-cell antigens (CD19, CD20, and CD22), by chimeric antigen receptor (CAR) T-cell [7,8,9] or bispecific T-cell engagers (BTCEs) [10], have provided new treatment options and strong improvement of patient outcome, effective immunotherapeutic agents still lack for T-ALL. Indeed, the development of such therapeutics has been hampered by the shared expression of many targetable antigens, such as CD2, CD3, CD5, and CD7 among neoplastic and normal cells, whose binding is predicted to produce T-cell fratricide effects and severe immunosuppression [11]. Therefore, it is conceivable that the targeting of T-ALL selective antigens might provide relevant therapeutic options for the management of this aggressive disease. With this aim, we recently reported the targeting of a unique CD43-associated glycol-epitope (UMG1) with a novel CD3ε-BTCE, as a safe and highly selective strategy against T-ALL [12]. This UMG1/CD43-CD3ε BTCE is presently under advanced preclinical development for clinical translation purposes in a first-in-human study.

In parallel, we investigated and present here UMG2/CD1a as a potential specific T-ALL target. CD1a is a glycoprotein expressed on ~40% of cortical-derived T-ALL patients and on an evolutionarily transient thymocyte population but not on peripheral blood cells [13]. On non-hematopoietic cells, only Langerhans cells (LC) express the antigen [14]. On the basis of this CD1a restricted pattern of expression and selective T-ALL target, a recent in vitro and in vivo study with anti-CD1a CAR-T cells showed promising anti-leukemia effects [15]. The efficacy, feasibility, and safety of anti-CD1a CAR-T is currently under investigation with the aim to move into the clinical setting.

With the aim to build a therapeutic platform against T-ALL, we developed a novel asy metric BTCE that simultaneously binds a unique epitope of CD1a and CD3ε (CD1a x CD3ε) to recruit and trigger a powerful T-cell-mediated anti-tumor response. We here report the in vitro and in vivo activity of this BTCE, supporting the translational value of CD1a-targeting strategies as a novel therapeutic tool for patients affected by this aggressive and orphan disease.

## 2. Materials and Methods

### 2.1. Cell Lines

PF-382, TALL-1, HPB-ALL, and Jurkat were purchased from DSMZ. CCRF-CEM and HEK293T cell lines were obtained from ATCC.

PF-382, TALL-1, CCRF-CEM, and Jurkat, were cultured in RPMI 1640 (Gibco, Thermo Fisher Scientific, Waltham, MA, USA), supplemented with 10% fetal bovine serum (Lonza Group Ltd., Basel, Switzerland), 100 U/mL penicillin, and 100 μg/mL streptomycin (Gibco, Thermo Fisher Scientific), and maintained at 37 °C in a 5% CO_2_ atmosphere.

The HPB-ALL cell line was cultured in RPMI 1640 supplemented with 20% fetal bovine serum.

The HEK293T cell line was cultured in DMEM (Gibco, Thermo Fisher Scientific) supplemented with 10% fetal bovine serum, 100 U/mL penicillin, and 100 μg/mL streptomycin.

### 2.2. Patient Samples

T-ALL cells were collected from Pediatric Hematology and Oncology Unit “Azienda Ospedaliera Pugliese-Ciaccio”, in compliance with bio-ethical standards. PBMCs were isolated by Ficoll-Paque Plus (Cytiva Europe GmbH, Buccinasco (Milan), Italy) density gradient centrifugation and then washed twice in culture medium (RPMI-1640 supplemented with 10% FBS).

### 2.3. Transfection of HEK293 Cells with an Expression Vector Encoding for CD1a

HEK293T cells were transiently transfected with the plasmid pLv203-GFP-puro vector encoding for CD1a (or pLv203-GFP-puro empty vector (Genecopoeia) by using Lipofectamine 3000 (Thermo Fisher Scientific, Waltham, MA, USA) according to the manufacturer’s instructions. Forty-eight hours after transfection, cells were subjected to puromycin selection at 0.5 μg/mL. Once selected, cells were assessed for positivity to UMG2/CD1a by FACS analysis.

To generate HEK293 cells stably expressing CD1A, 500 μL of CD1A (or empty vector-EV-) lentiviral particle containing supernatant were used for transduction of 2.5 × 10^5^ cells plated in a six-well plate one day before transduction. Cell transduction was performed with an over-night incubation with lentiviral supernatant in a cell culture incubator at a final volume of 2 mL of HEK293T cell media in the presence of polybrene at the final concentration of 8 μg/mL. Forty-eight hours after transduction, cells were subjected to puromycin selection at 0.5 μg/mL. After antibiotic selection, cells were assessed for the expression of CD1a before proceeding with the cytotoxicity assay.

### 2.4. CD1a Lentivirus Production

Furthermore, 5 × 10^6^ HEK293T were co-transfected with 3 μg of lentiviral pLv203-GFP-puro vector encoding for CD1a (or pLv203-GFP-puro empty vector) and 9 μg of an optimized mixture of the three packaging plasmids pLP1, pLP2, and pLP/VSVG (ViraPower Lentiviral Packaging Mix, K497500, Thermo Fisher Scientific) by using Lipofectamine LTX (A12621, Thermo Fisher Scientific) according to manufacturer’s protocol. Forty-eight hours post-transfection, lentivirus containing supernatant was harvested, filtered using 0.45 μm sterile filter, aliquoted, and stored at −80 °C for further experiments.

### 2.5. UMG2 and Anti-CD1a Binding Reactivity

UMG2 reactivity against healthy donor peripheral blood cells, HEK293T and T-ALL cell lines was evaluated by flow cytometry. Cells were incubated with UMG2 unconjugated mAb and then stained with Alexa Fluor 647 or PE goat anti-mouse IgG secondary antibody (Jackson ImmunoResearch Europe Ltd., Ely, UK).

CD1a reactivity was evaluated on T-ALL cell lines. Cells were incubated with anti-human CD1a PE commercial MAbs: anti-human CD1a PE clone SK9 (Biolegend, San Diego, CA, USA), anti-human CD1a PE clone HI149 (Biolegend), anti-human CD1a PE clone BL6, (Beckman Coulter, Brea, CA, USA), and anti-human CD1a PE clone NA1/34-HLK (Invitrogen, Thermo Fischer Scientific, Waltham, MA, USA).

Multidimensional flow cytometry immunophenotyping of healthy donor peripheral blood cells was performed using the following anti-human monoclonal antibodies: anti-CD45 BV510 (BD Biosciences, San Jose, CA, USA), anti-CD3 PerCP-Cy5.5 (BD Biosciences), anti-CD4 FITC (BD Biosciences), anti-CD8 APC-H7 (BD Biosciences), anti-CD19 PECy-7 (BD Biosciences), anti-CD16 BV421 (BD Biosciences), UMG2 mAb, and-Alexa Fluor 647 goat anti-mouse IgG secondary antibody (Jackson ImmunoResearch, Ely, United Kingdom).

Competitive binding assay was performed through FACS analysis by using unconjugated UMG2 mAb, UMG2 PE-conjugated, and the following anti-human PE fluorochrome conjugated CD1a mAbs: anti-CD1a PE clone SK9, (Biolegend), anti-CD1a PE clone HI149 (Biolegend), anti-CD1a PE clone BL6 (Beckman Coulter), and anti-CD1a NA1/34-HLK PE (Invitrogen). HPB-ALL cells were incubated for 30 min at 2–8 °C in the dark with unconjugated UMG2 mAb at increasing volumes (0.01 μL, 0.1 μL, 0.5 μL, 1 μL, 2 μL, 5 μL) in the presence of saturation concentration of each CD1a mAb or UMG2 PE-conjugated (positive control). Isotype matched fluorochrome-conjugated antibodies were used as control to establish background staining. One hundred thousand cells were stained for each test. After incubation, samples were washed in PBS and acquired on an Attune NxT flow cytometer (Thermo Fisher Scientific). All data were analyzed by Attune NxT Flow Cytometer Software.

### 2.6. RNA Expression Level of CD1a on Primary Samples

RNA expression data of 10 different cell types isolated from primary human bone marrow and thymus were obtained from the GSE69239 data set. The expression profiling was performed by high throughput sequencing (Illumina HiSeq 2000). Expression was estimated in units of FPKMs (fragments per kilobase of mappable gene length and million reads). Technical information has been deeply described by David Casero et al. [16]. We calculated the relative RNA expression level of CD1A, by normalizing FPKMs of CD1A respect to FPKMs referred to SDHA, that has been used as reference gene.

We calculated the relative RNA expression level of CD1a, by normalizing FPKMs of CD1Arespect to FPKMs referred to SDHA, that has been used as reference gene.

### 2.7. Generation of UMG2-BTCEs

CD1a/UMG2 and the ScFv of anti-hCD3 clone L2K-07 cDNAs were cloned into Evitria’s vectors using non-PCR based cloning techniques. The vector plasmids were synthesized. Plasmid DNA was prepared under low-endotoxin conditions based on anion exchange chromatography. DNA concentration was determined by measuring the absorption at 260 nm. Correctness of the sequences was verified with Sanger sequencing. Suspension-adapted CHO K1 cells (originally received from ATCC) were used for the construct production. The seed was grown in eviGrow medium, a chemically defined, animal-component-free, serum-free medium. Cells were then transfected with eviFect and were grown after transfection in eviMake2. Supernatant was harvested by centrifugation and subsequent filtration (0.2 µm filter). The antibody was purified using MabSelect SuRe. Monovalent CD3 binding BTCE was generated using “Knobs-into-holes” technology. Purity was determined by analytical size exclusion high performance chromatography (HPLC) with an Agilent AdvanceBio SEC column (300 A 2.7 μm 7.8 × 300 mm) and DPBS as running buffer at 0.8 mL/min. At the end of this process, 42 mg of purified CD1a x CD3ε were obtained and resuspended in PBS 100 mmol/L L-arginine at the concentration of 4.63 mg/mL. The final product was characterized by 36% monomoraicity and <1 EU/mg of endotoxin.

### 2.8. Kd Calculation

Kd was determined through FACS analysis. For this assay, HPB-ALL were resuspended in RPMI 1640 with 10% FBS and 1% P/S, the cell concentration was adjusted at 1 × 10^6^ cell/mL. One hundred thousand cells per tube were incubated with increasing concentrations of CD1a x CD3ε (0.001 μg/mL, 0.01 μg/mL, 0.1 μg/mL, 1 μg/mL, 10 μg/mL, 100 μg/mL). Cells were stained for 30 min at 2–8 °C. HPB-ALL cells were washed with PBS and centrifuged for 5 min at 1300 rpm. Cells were stained with PE fluorochrome conjugated secondary antibody for 30 min at 2–8 °C. HPB-ALL cells were washed as previously described and resuspend in PBS for flow cytometry analysis. After acquisition, median fluorescence intensity (MFI) was determined, and Kd was calculated using GraphPad Software (La Jolla, CA, USA).

### 2.9. Redirected T Cell Cytotoxicity Assay

T-ALL cell lines, HEK293T EV, HEK293T/CD1a, and primary T-ALL cells were labeled with Far-Red (Thermo Fisher Scientific) viable marker, according to manufacturer instructions. Labeled cells were co-cultured with PBMCs at different E:T ratios, in the presence of increasing concentrations of CD1a x CD3ε or vehicle for 24–48 h at 37 °C and 5% CO_2_, and then stained with 7-AAD (BD Biosciences). Cytotoxicity was detected by flow cytometry (Attune NxT Flow cytometer, Thermo Fisher Scientific). To identify dead T-ALL cells, the following gating strategy was used: 7-AAD−/Far Red+ cells were considered as alive T-ALL cells while 7-AAD+/Far Red+ were considered dead T-ALL cells. Moreover, 7-AAD−/Far Red− were considered as alive PBMC while 7-AAD+/Far Red− were considered dead PBMC. In the cytotoxicity experiment with T cell depletion, immunomagnetic cell sorting using CD4, CD8, or CD56 microbeads (MACS Miltenyi Biotec, Bergisch Gladbach, Germany) was performed. In the cytotoxicity experiment with Fc blocking, Fc receptor binding inhibitor antibody (Invitrogen, Waltham, MA, USA) was added to cell co-culture according to manufacturer instructions.

### 2.10. Direct Cytotoxicity Assay

T-ALL and HEK293T cells were plated in the absence of effector cells in a 96 well plate, treated with increasing concentrations of CD1a x CD3ε or vehicle and incubated at 37 °C and 5% CO_2_. After 72 h, 10 μL of CCK8 solution (Dojindo Molecular Technologies, Inc., Rockville, MD, USA) was added to the cell culture, cells were incubated for a further 4 h, and the absorbance at 450 nm was measured using a microplate reader.

### 2.11. T Cell Activation and Proliferation

T-ALL target cells were plated as is in cytotoxicity assays and were incubated with PBMCs at increasing E:T ratios for 24–48 h at 37 °C and 5% CO_2_. T cells were stained with anti-human CD4 PE-Cy7, CD8 APC-Cy7, CD25 APC, CD69 PE/APC (BioLegend/BD Biosciences), and CD107a APC (BD Biosciences, 4 h of incubation). T cells were differentiated from T-ALL cells using different parameters, including size and granularity (complexity) by gating on FSC-A SSC-A flow cytometry and then selected as Far-Red-negative, gated for CD4 or CD8, and for CD69, CD25, or CD107a. For IFN-y, TNF-a, perforin, granzyme, granulysin, and IL-2 intracellular staining, T-ALL target cells and PBMCs were plated at 10:1 E:T ratio in the presence of increasing concentration of CD1a x CD3ε or negative control. Brefeldin A 10 μg/mL was added, and cells were incubated at 37 °C, 5% CO_2_ for 4 h.

Target and effector cells were washed twice with room temperature 1× PBS and subsequently stained for 20 min with CD4 PE-Cy7 and CD8 APC-H7 (BD Biosciences).

Cells were washed twice with 1× PBS, fixed with reagent A (Nordic-MUbio, Susteren, The Netherlands) for 15 min protected from light, then washed with 1× PBS, permeabilized with Reagent B (Nordic-MUbio), and stained with anti-TNF-a PE (BD Biosciences), anti-IFN-y BV-421 (BD Biosciences), anti-granulysin AF-488 (BD), anti-granzyme AF 647 (BD Biosciences), anti-perforin PECF594 (BD Biosciences), and anti-IL-2 PE-Cy7 mAbs (Thermo Fisher Scientific) for 15 min at RT protected from light. After incubation, samples were washed in 1× PBS and analyzed by ATTUNE NxT flow cytometer (Thermo Fisher Scientific).

T Cell proliferation was measured using Cell Trace Violet (Thermo Fisher Scientific) staining and Cell Trace Violet dilution analysis. Briefly, Cell Trace Violet stained PBMCs were co-cultured with T-ALL cell lines as previously described for cytotoxicity assay. After 5 days of co-culture, cells were washed with PBS and acquired immediately by flow cytometry (ATTUNE NxT, Thermo Fisher Scientific).

### 2.12. Immunofluorescence Microscopy

T-ALL cell lines were stained with Cell Trace Far Red and counterstained by Hoechst 33342 (1 μg/mL in dH_2_O), and were then co-cultured with PBMCs at 10:1 E:T ratio for 48 h in the presence of vehicle or 0.1 μg/mL CD1a x CD3ε. After 48 h of co-culture, cells were washed and stained with anti-CD3 FITC, then washed twice with 1× PBS and spotted by cytospin on microscope slides. Cells were then fixed by 4% PFA in PBS and finally wet-mounted for immunofluorescence imaging.

### 2.13. CD1a x CD3ε + Immune Checkpoint Inhibitors (ICIs) Combinatorial Approach

To recapitulate chronic antigen stimulation in vitro, freshly isolated PBMCs derived from healthy donors were co-cultured at 10:1 E:T ratio, as previously described for cytotoxic assay, with Cell Trace Far Red labeled CD1a+ T-ALL cells in the presence of CD1a x CD3ε alone, nivolumab alone, avelumab alone, combination of CD1a x CD3ε + nivolumab and CD1a x CD3ε + avelumab, for 5 days. Cell Trace Violet labeled CD1a+ T-ALL cells and drugs were then added to the culture. After 5 days, T cell exhaustion markers were evaluated by multiparametric flow cytometry panel (CD4-FITC, CD8-APC-H7, CD69-PE, PD-1-PE-CF594, TIM3-BV421, TIGIT-BV711, PD-L1 FITC BD Biosciences) and cytotoxicity was assessed by flow cytometry (ATTUNE NxT, Thermo Fisher Scientific).

### 2.14. In Vivo Studies

In vivo experiments were approved by the National and Institutional Animal Committee. All the procedures were performed according to standard guidelines and approved protocols. Four-to-six-week-old female NSG (NOD.Cg-PrkdcscidIl2rgtm1Wjl/SzJ) mice were purchased from Charles River Laboratories (Wilmington, MA, USA). During experiments, animals were regularly monitored and euthanized when signs of disease-related symptoms or graft-versus-host disease (GvHD) developed.

Furthermore, 5 × 10^6^ HPB-ALL cells were intravenously injected. At day 7, 20 × 10^6^ PBMCs were injected into the tail vein of each mouse. Three days after PBMCs engraftment, mice were randomized into 3 groups (cohorts of 5 animals) of treatment: (1) vehicle, (2) 0.1 mg/kg CD1a x CD3ε, and (3) 0.5 mg/kg CD1a x CD3ε and mice were intraperitoneally (ip) injected. Tumors were monitored with IVIS LUMINA II Imaging System (Caliper Life Sciences, Waltham, MA, USA) after 4 h from tail vein injection of RediJect 2-DeoxyGlucosone (2-DG) (PerkinElmer, Waltham, MA, USA).

### 2.15. Statistical Analysis

Each in vitro experiment was performed at least 3 times. Values are expressed as means ± SD/SEM. Statistical evaluation between two group differences were performed with Student’s *t*-test by GraphPad software (www.graphpad.com, access on 9 May 2022). Graphs were obtained using Graphpad Prism version 6.0. Effects were considered statistically significant when *p* value was less than 0.05.

## 3. Results

### 3.1. Generation and Characterization of Humanized UMG2 mAb

By long-term culture and several subcloning procedures, we selected the UMG2 clone from a previous generated murine hybridoma [17] that has been characterized and subsequently clustered as anti-CD1a mAb [18]. While the epitope recognized by UMG2 mAb is expressed by CD1a, the affinity against the combinatory site is slightly improved when compared to the original parental clone. This clone was then sequenced for the generation of an asymmetric bispecific construct with a CD3ε monovalent arm.

To confirm that UMG2 mAb recognizes a CD1a epitope, the HEK293T cell line, that does not express CD1a endogenously, was transiently transfected with a plasmid encoding for CD1a or with an empty vector (EV) as negative control (Figure 1A). After puromycin selection, the specific binding of UMG2 or other anti-CD1a (SK9, BL6, and HI149) mAbs to CD1a were assessed by flow cytometry. A strong binding of UMG2 mAb to CD1a-expressing HEK293T was found, while no reactivity was observed on HEK293T cells transfected with the negative control (Figure 1B). Importantly, a similar pattern of reactivity was observed for the other anti-CD1a mAbs tested, further confirming the reactivity of UMG2 mAb towards CD1a (Appendix A). Next, UMG2 and clinical validated anti-CD1a mAb (NA1/34-HLK) reactivity was evaluated on a panel of T-ALL established cells (Figure 1C). Moreover, to investigate if UMG2 mAb binds an original CD1a epitope, competitive binding assays between UMG2 and fluorochrome-conjugated SK9, BL6, HI149, or NA1/34-HLK anti-CD1a mAbs were performed. Interestingly, none of the SK9, BL6, and HI149 mAbs competed with UMG2 mAb binding, while a partial competition between NA1/34-HLK and UMG2 was observed (Figure 1D, Appendix A). Furthermore, anti-UMG2 staining on primary T-ALL cell was performed. One out of 3 primary T-ALL samples tested was cortical (EGIL T-III) and, consistently, a strong reactivity of UMG2 was found (Figure 1E). Moreover, UMG2 mAb reactivity was evaluated on PBMCs from healthy donors. No reactivity was found on different blood cell subtypes as expected (Figure 1F), consistently with the restricted pattern of expression of CD1a.

Finally, in order to evaluate the expression level of CD1a across primary human bone marrow and thymic (Thy) progenitor cells, we queried a public available dataset (GSE69239) including RNA-sequencing data from 10 distinct cell types isolated from bone marrow and thymus (Figure 1G). We found that Thy samples express CD1a at different transcriptional levels depending on the commitment stage. Fully T cell committed populations CD4 + CD8 + (indicated as Thy 4) express CD1a at higher transcriptional level with respect to Thy 3 (CD34 + CD7 + CD1a + CD4negCD8neg) and Thy5 (CD3 + CD4 + CD8neg) (intermediate level). Thy1 (CD34 + CD7neg CD1aneg CD4negCD8neg), Thy2 (CD34 + CD7 + CD1aneg CD4negCD8neg), and Thy 6 (CD3 + CD4neg CD8+) present a low or negative transcriptional level. No expression has been detected on hematopoietic stem cells (HSC), lymphoid-primed multipotent progenitors (LMPP), common lymphoid precursor (CLP), and fully B cell committed progenitors (BCP).

### 3.2. Structural Characteristics and Binding Properties of CD1a x CD3ε

We designed a novel anti-CD1a BTCE with a monovalent CD3ε binding site to reduce unspecific T-cell triggering in the absence of concomitant CD1a binding. The domain to CD1a has been designed with a bivalent arm to empower targeting avidity to antigen-expressing T-ALL cells. To this end, a novel asymmetric (2 + 1) CD1a x CD3ε was generated (Figure 2A,B).

To define the BTCE apparent constant of dissociation (Kd), dose escalating experiments were performed. Average apparent Kd was estimated at 0.014 µg/mL, while the binding saturation was reached at concentrations of about 1 µg/mL (Figure 2C).

### 3.3. CD1a x CD3ε Recruits and Triggers T-Cell Mediated Cytotoxicity against CD1a-Expressing T-ALL Cells

The in vitro activity of CD1a x CD3ε was first evaluated using HEK293T EV and HEK293T stably expressing CD1a cells, co-cultured with PBMCs at 10:1 effector to target (E:T) ratio. Importantly, concentration-dependent T-cell redirected cytotoxicity occurred against HEK293T/CD1a but not towards HEK293T EV cells (Figure 2D). To investigate if CD1a x CD3ε induced direct cytotoxic effect in the absence of T-cells, HEK293T/CD1a and HEK293T/EV cell viability was assessed after 72 h of treatment with CD1a x CD3ε alone. As expected, no differences in cell viability were observed between HEK293T/CD1a and HEK293T cells (Appendix A).

Next, to validate the translational relevance of these findings, CD1a-expressing and not expressing T-ALL cells were co-cultured with PBMCs at 10.1 E:T ratio in the presence of increasing concentration of CD1a x CD3ε. The treatment resulted in strong cytotoxic effect against CD1a-expressing T-ALL cells (Figure 2E,F) but not against CD1a negative cells (CCRF-CEM). Importantly, CD1a x CD3ε induced T-cell redirected cytotoxicity leading to about 60% killing of CD1a-expressing primary T-ALL cells (Figure 2G). Consistent with the absence of CD1a expression on the surface of normal peripheral blood mature T lymphocytes, no fratricide cytotoxic effects were observed on primary T cells exposed to increasing concentrations of CD1a x CD3ε (Appendix A). Moreover, to investigate the in vitro effectiveness of the CD1a x CD3ε in the context of active T-ALL disease, a cytotoxic assay at low E:T ratio (from 1:1 to 5:1) was performed by co-culturing PBMCs with HPB-ALL cell line in the presence of CD1a x CD3ε. Notably, the BTCE was very active at low E:T ratio (Appendix A), suggesting a potential role of CD1a x CD3ε in T-ALL active disease. In addition, concentration-dependent CD107a expression increased in T lymphocytes co-cultured at a 10.1 E:T ratio with both T-ALL cell lines and primary T-ALL cells, indicating CD1a x CD3ε-induced degranulation (Figure 2H,I).

### 3.4. CD1a x CD3ε Triggers T-Cell Activation against T-ALL Cells

To demonstrate CD1a x CD3ε-driven T-cells activation, CD1a+ (HPB-ALL, TALL-1, and Jurkat) and CD1a- (CCRF-CEM) T-ALL cell lines were co-cultured with PBMCs at 10.1 E:T ratio in the presence of increasing concentration of CD1a x CD3ε. Importantly, CD1a x CD3ε led to concentration-dependent T lymphocyte activation, as evaluated by up-regulation of CD69 and CD25 on both CD4 and CD8 T lymphocytes co-cultured with CD1a+ T-ALL, as well as by the release of granzyme, perforin, and granulysin, and the production of pro-inflammatory cytokines, such as TNF-α, IFN-γ, and IL-2 (Figure 3A–C). Notably, a negligible level of T-cell activation was found when PBMCs were co-cultured with CD1a- T-ALL cell line in the presence of increasing concentrations of CD1a x CD3ε (Appendix A).

Finally, to demonstrate that the observed CD1a x CD3ε-induced cytotoxic effect is indeed T lymphocyte-dependent (Figure 4A), CD1a + T-ALL cells were exposed to increasing concentrations of CD1a x CD3ε and co-cultured with total PBMCs, CD4 depleted, or CD8 depleted PBMCs. Importantly, minimal cytotoxic activity was observed in both CD4 and CD8 depleted samples as compared with total PBMCs (Figure 4B). In addition, to exclude that the presence of the Fc domain in the CD1a x CD3ε construct could induce the recruitment of natural immunity effector cells through Fc-FcγR interaction leading to cytokine release syndrome, the mechanisms of action of CD1a x CD3ε were further investigated. Jurkat cells were co-cultured at 10.1 E:T ratio with undepleted PBMCs, CD56 depleted, total PBMCs exposed to Fc blocker, or only CD56+ lymphocytes. Whereas minimal activity was found in the presence of CD56+ lymphocytes alone, both CD56 depleted, and Fc blocked total PBMCs were able to achieve cytotoxic activity comparable to total undepleted PBMCs (Figure 4C). Furthermore, no direct cytotoxicity was observed in the absence of effector cells, again the treatment with CD1a x CD3ε did not trigger direct cytotoxic effect. (Figure 4D). Moreover, CD1a x CD3ε led to concentration-dependent proliferation of T cells co-cultured with CD1a expressing cell lines (Figure 4E).

Taking these results together indicates that CD1a x CD3ε exerts T-cell mediated cytotoxicity against CD1a-expressing T-ALL cells which does not involve activation of other cytotoxic cells.

### 3.5. CD1a x CD3ε Is Highly Effective against T-ALL Xenografts in a NOD-SCID Mice Model Reconstituted with the Human PBMC

The in vivo anti-tumor activity of CD1a x CD3ε against CD1a-expressing T-ALL cells was investigated. HPB-ALL cells were xenografted into non-obese diabetic/severe combined immunodeficiency NOD *scid* gamma chain null mice (NSG). Seven days later, human PBMCs derived from a healthy donor were injected into the animals. Then, 3 days later, mice were randomized to receive CD1a x CD3ε bi-weekly ip (0.1 mg/kg or 0.5 mg/kg) or vehicle. Tumor growth was evaluated with a fluorescent IVIS imaging probe. (Figure 5A). As compared with the untreated control group, mice treated with 0.1 or 0.5 mg/kg CD1a x CD3ε showed significantly reduced tumor growth at both concentrations (Figure 5B) and this effect translated into a prolonged survival of animals with a median survival of 41 days in vehicle group and 62 and 63 days in mice treated with 0.1 or 0.5 mg/kg CD1a x CD3ε, respectively (Figure 5C).

### 3.6. CD1a x CD3ε and ICIs Combinatorial Approach

Chronic antigen stimulation is known to produce up-regulation of PD-L1/PDL-2 on tumor cells and expression of exhaustion markers, such as PD-1, TIM-3, LAG-3, and TIGIT on T-lymphocytes, which hamper immune response [19]. We reasoned that ICIs combined with CD1a x CD3ε may counteract T-cell dysfunction offering a promising field of investigation.

To this end, PBMCs and T-ALL Cell Trace Far Red labeled cells were co-cultured in the presence of CD1a x CD3ε or vehicle. After 72 h, T lymphocytes were restimulated with T-ALL Cell Trace Violet labeled cells at 10:1 E:T ratio in the presence of CD1a x CD3ε plus anti-PD-1 or plus anti-PD-L1, or vehicle. We found a significant reduction in expression of T-cell exhaustion markers on T lymphocytes co-cultured with CD1a-expressing cells in the presence of CD1a x CD3ε plus anti-PD-1 or anti-PD-L1 (Figure 6) without increased toxicity in the short-term assays (data not shown). This preliminary in vitro finding suggests that in contrast to other malignancies, bispecific mAbs plus ICIs may not represent an effective strategy in T-ALL [20]. However, in vivo data should be required to definitively solve this important question.

## 4. Discussion

To date, the cure of refractory/relapsed T-ALL patients is still an unmet need. While novel approaches have revolutionized the perspectives of the B-ALL patients, no immunotherapy-based strategies are presently approved for T-ALL.

Here, we report the generation and preclinical validation of a novel tool for the targeting of a previously uncharacterized CD1a epitope. Consistent with the pattern of expression of CD1a, we found that the novel UMG2 mAb strongly binds cortical-derived T-ALL cells, but not healthy peripheral blood cells, confirming a safe profile of expression of the epitope recognized. On these premises, we engineered a novel BTCE, the CD1a x CD3ε, to investigate the translational relevance of redirecting T-cell response against CD1a-expressing T-ALL cells.

BTCEs are emerging immunotherapeutic tools which rely on the promotion of an immune synapse between a tumor-associated antigen (TAA) and immune effector activating antigens. Therefore, BTCEs are constructs with a cell-bridging function not present in any combination of parent antibodies [21]. There are different formats of BTCEs, ranging from very small proteins, consisting of two single chain variable fragments (scFv), to larger asymmetric or symmetric immunoglobulin G (IgG)-like molecules [21].

Our construct is a complete IgG-scFv bispecific antibody composed of two arms belonging to UMG2 mAb binding CD1a with an antibody-backbone linked to a single scFv, derived from anti-CD3 mAb L2K-07. This asymmetric format (2 + 1) was selected for advantageous pharmacodynamic and pharmacokinetic reasons. First, the presence of bivalent binding domains for CD1a increases the avidity and selective recognition of CD1a antigen-expressing cortical T-ALL cells [22]. Second, the monovalent arm for CD3ε is preferred to a bivalent binding to reduce unspecific T cell activation by CD3 cross-linking and also to minimize BTCE trapping in normal tissues rich in T lymphocytes [23]. Third, the presence of neonatal Fc receptor (FcRn) protects IgG-like BTCE from rapid degradation and confers long plasma half-life (days) as compared to the shorter plasma half-life (hours) of Fc-fragment lacking BTCE that, instead, need continuous infusion [21,24,25].

Notably, in vitro CD1a x CD3ε produced a concentration-dependent activation of T-cells, release of inflammatory cytokines, and induction of T-cell proliferation and activation, leading to T-ALL cells lysis in a dose- and E:T ratio-dependent manner. Consistently with these in vitro data, CD1a x CD3ε led to a significant anti-tumor activity and survival advantage of treated animals in a mouse model of human T-ALL reconstituted with human immune effectors. Importantly, for the translational value of CD1a x CD3ε, we found this activity in a range of concentrations similar to the doses used for the first clinically approved BTCE, blinatumomab [26].

Our data could be of clinical relevance given the urgent need for novel therapeutics for T-ALL [27,28,29]. Indeed, the development of immunotherapeutic agents in this aggressive disease has been mostly impaired by the shared expression of targets between normal and neoplastic cells, whose targeting may produce severe T-cell aplasia and life-threatening opportunistic infections [28,30], making the treatment highly toxic for frail relapsed/refractory patients. The ideal target, in fact, should be highly expressed by malignant cells, not detectable on normal mature peripheral blood T-cells to avoid the “fratricide” activity, and not expressed by non-hematopoietic tissues to avoid treatment-limiting on-target off-tumor effects. Thus far, different T-cell antigens have been investigated with this aim. CD7 is a pan-T antigen widely expressed during T differentiation stages from progenitors to mature T/NK cells, and steadily expressed on the majority of T-ALL, making its targeting at risk of profound T-cell aplasia and intensive supportive care [31,32]. Similarly, CD5 is significantly expressed on T-ALL, except for ETP-ALL, but also widely expressed by mature T-cells, therefore limiting its use due to a risk of deep immunosuppression following its targeting. Moreover, CD5 is rapidly internalized. This event, while reducing the fratricide effect, also represents a potential escape mechanism for tumor cells [28]. CD38 is presently under investigation as a potential target for T-ALL. It is steadily expressed by T-ALL cells at presentation and primary refractory and relapsed/refractory stages [33], but also widely expressed within cells of the hematopoietic compartment and non-hematopoietic compartments, including the smooth muscle and pancreatic islets [34]. Even if the anti-CD38 daratumumab has been approved in multiple myeloma patients [35] and preclinically assessed against T-ALL [36], a careful clinical monitoring is required for on-target off-tumor toxicities in these frail patients. At present, consistent clinical data are eagerly awaited from ongoing studies (NCT04785833, NCT04620655, NCT04582487). A potential target of interest is also the anti-CD96 mAb (TH-111) that has been shown to stain a major (78.3%) subset of T-ALL [37] with limited reactivity with blood and bone marrow cells, therefore representing a potential target. However, it is still waiting for clinical translation [38]. CC chemokine receptor 4 (CCR4) is highly expressed in adult T-cell leukemia-lymphoma (ATL) cells, and a monoclonal antibody against CCR4, induced significant killing of ATL cells via antibody-dependent cellular cytotoxicity (ADCC). However, serious adverse cutaneous reactions, such as Stevens–Johnson syndrome, have been reported, thus also hampering its use in T-ALL patients [39,40].

Alemtuzumab is a humanized monoclonal antibody anti-CD52, which is expressed on most malignant T-lymphocytes. Indeed, alemtuzumab showed promising activity in post-thymic, T-cell malignancies, including T-cell prolymphocytic leukemia (T-PLL), cutaneous T-cell lymphoma (CTCL), T-cell large granular lymphocyte (T-LGL) leukemia, and human T-cell lymphotropic virus I (HTLV-I)-associated adult T-cell leukemia-lymphoma (ATLL) [41]. However, no responses were observed in T-ALL patients, and the shared expression of CD52 among normal and malignant T-lymphocytes, expose patients to dangerous effects of lymphopenia, such as reactivation of cytomegalovirus, herpes simplex, or herpes zoster-latent infections [42].

Very recently, a new target for T-ALL was proposed by our group [12]. A novel unique epitope of CD43, recognized by mAb UMG1, was found to be expressed by T-ALL. Indeed, among 110 T-ALL samples, approximately 50% of cases highly expressed the target (82% of EGIL TIII patients), while <5% of mature T peripheral blood lymphocytes expressed the epitope, significantly excluding the fratricide risk. Moreover, a wide screening of normal tissues, including vital organs, excluded target off-tumor effects, demonstrating a safe expression profile and making this epitope a relevant candidate for targeting in a clinical setting [12].

CD1a is a surface glycoprotein expressed on approximately 40% of T-ALL cases, where it defines a cortical-derived T-ALL subgroup. CD1a expression is associated with an excellent clinical outcome; however, about 33% of relapsed/refractory patients still express CD1a [33]. In this regard, CD1a x CD3ε could provide an effective option for a subset of patients with poor therapeutic resources and dismal clinical outcome.

On normal tissues, CD1a is only expressed by a subset of skin-resident dendritic cells (Langerhans cells, LC) and on a transient thymocyte population, but not on mature T cells. This pattern of expression makes the targeting of CD1a fratricide-free and with limited on-target off-tumor effects supporting the idea of a personalized immunotherapeutic option for that sub-population of T-ALL patients, even rarely expressing CD1a [15]. Consistent with its promising features as selective T-ALL therapeutic target, recently an anti-CD1a CAR-T was developed, showing significant in vitro and in vivo anti-tumor effects.

In this light, the targeting of CD1a by a novel BTCE offers a concrete option, with some aspects that make it more advantageous when compared to CAR-Ts. Indeed, it must be considered that BTCE-based therapy is an effective off-the-shelf strategy that does not require the complex and expensive ex vivo manipulation of effector cells for CAR-T generation. Furthermore, BTCEs are characterized by a better dose management, which reduces the risks associated with the cytokine-release syndrome (CRS) and other toxicities commonly associated with CAR-T therapy [43].

In our data, the targeting of CD1a may cover cases that are CD43/UMG1-negative, therefore providing an experimental platform for the treatment of T-ALL by this BTCE. Furthermore, our data showed that UMG2 epitope pattern of expression is more restricted as compared to most other CD1a mAbs but similar to NA1/34-HLK that was recently selected for clinical use. The difference in specificity of the epitope recognized by UMG2 as compared to most available mAbs against the same target may be due to post-transcriptional modifications of CD1a, a highly glycosylated surface protein, generating the specific epitope. This can be of translational relevance because the epitope expression may correlate with different T-ALL biologic features and prognoses, and therefore could drive a better patient selection in future precision immunotherapy platforms. Moreover, we do not expect life-threatening side effects from targeting UMG2 epitope expressing normal thymocytes, due to their marginal role in adult life, and also because T-cells move from the thymus very early in the embryonic life [44]. For these reasons, we think that our approach does not confer the risk of immunodeficiency derived from T-cell repertoire impairment, which is potentially produced by other highly shared antigens among normal and neoplastic T lymphocytes.

## 5. Conclusions

In conclusion, to our knowledge, our data support the feasibility and the efficacy of a novel BTCE targeting T-ALL cells, with a very low risk of immunosuppression, with the aim of a precision immunotherapy in these patients. We demonstrated that CD1a x CD3ε could represent an effective anti-T-ALL agent to be investigated in a first-in-human clinical trial as maintenance treatment, such as blinatumomab, for purging the minimal residual disease (MRD) or in CD1a-expressing refractory/relapsed patients to improve the poor disease control obtained with nelarabine. Furthermore, we hope that the present proof-of-concept study can provide a framework for CD1a x CD3ε incorporation within a chemo-free first-line treatment as a bridge to allogeneic stem-cell transplantation, therefore opening new opportunities for the treatment of this still incurable and aggressive orphan disease.

## Figures and Tables

**Figure 1 cancers-14-02886-f001:**
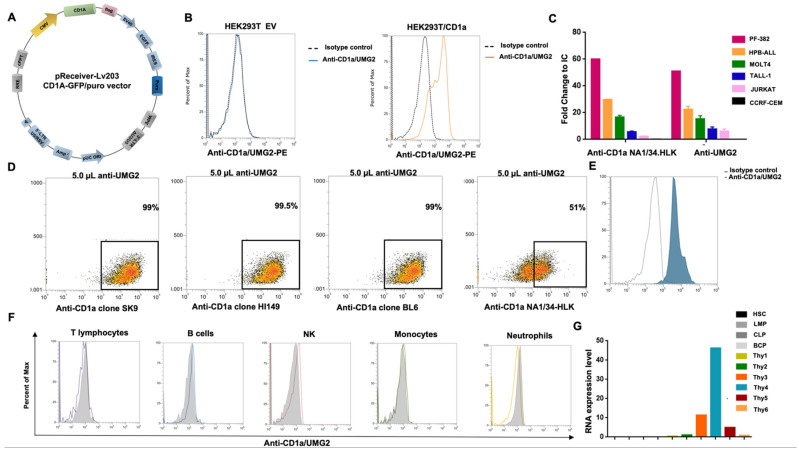
UMG2 mAb binding reactivity. (**A**) Graphical representation of CD1a/GFP vector map. (**B**) UMG2 reactivity on HEK293T cell line transfected with empty vector (left) or CD1a-expressing plasmids (right). (**C**) Reactivity of anti CD1a clone NA1/34-HLK and anti-UMG2 on T-ALL cell lines. (**D**) Competitive binding assay between unconjugated anti-UMG2 and commercially available anti-CD1a fluorochrome-conjugated antibodies (Bl6, SK9, HI14, and NA1/34-HLK) on HPB-ALL cell line. Only NA1/34-HLK anti-CD1a commercial clones partially compete with anti-UMG2. (**E**) Representative FACS data of UMG2 binding reactivity on primary T-ALL cells. (**F**) UMG2 binding on peripheral blood cells from healthy donors. (**G**) CD1a RNA expression level on human bone marrow and thymic progenitor cells.

**Figure 2 cancers-14-02886-f002:**
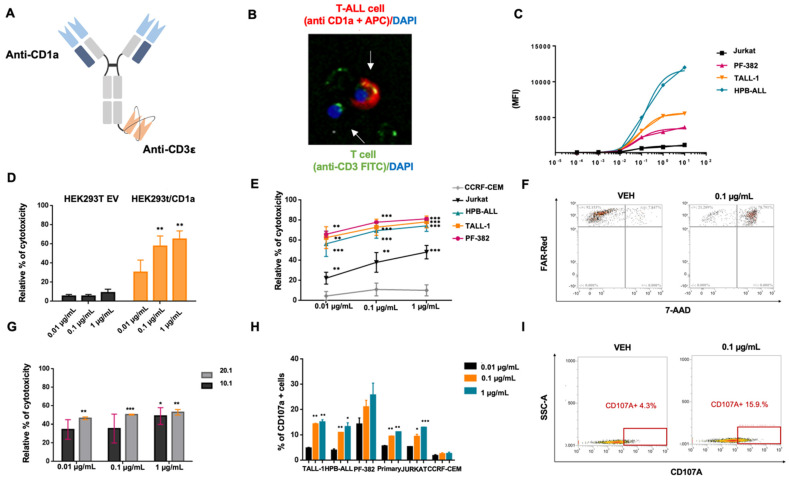
CD1a x CD3ε BTCE in vitro cytotoxic activity. (**A**) Schematic representation of CD1a x CD3ε BTCE. (**B**) Coupling of CD3+ T lymphocyte on CD1a+ T-ALL cell by CD1a x CD3ε BTCE assessed by immunofluorescence microscopy. CD3+ T cells are stained with DAPI (blue) and CD3 FITC (green), CD1a+ T-ALL cells are stained with DAPI (blue) and cell trace Far Red (red). (**C**) Average apparent CD1a x CD3ε BTCE Kd (0.014 µg/mL) evaluated on CD1a cortical derived T-ALL cell lines. (**D**) Relative percentage (%) of T cell redirected cytotoxicity of CD1a x CD3ε BTCE on HEK293T transduced with empty vector (black) and with a vector encoding for CD1a (orange). (**E**) Relative percentage (%) of T cell redirected cytotoxicity of CD1a x CD3ε BTCE against CD1a+ and CD1a- cortical derived T-ALL cell lines. (**F**) Representative FACS data (dot plots) of T cell redirected cytotoxicity on T-ALL cell line (TALL-1) co-cultured with PBMCs at a 10:1 E:T ratio in the presence of vehicle or 0.1 μg/mL of CD1a x CD3ε BTCE. (**G**) Relative percentage (%) of T cell redirected cytotoxicity of 0.1 μg/mL of CD1a x CD3ε BTCE against primary CD1a+ T-ALL cells (one representative patient derived sample). (**H**) CD107a concentration-dependent increase on T lymphocytes co-cultured at a 10.1 E:T ratio with both CD1a expressing T-ALL cell lines and primary T-ALL blasts (one representative patient derived sample). (**I**) Representative FACS data (dot plots) of CD107a increase on T lymphocyte co-cultured with T-ALL cells at a 10:1 E:T ratio treated with vehicle or 0.1 μg/mL of CD1a x CD3ε BTCE. * *p* < 0.05, ** *p* < 0.01, *** *p* < 0.001. *p* values are calculated by comparing co-cultures of PBMCs + T-ALL exposed to vehicle with co-cultures of PBMCS + T-ALL treated with increasing concentration of CD1a x CD3ε (0.01 µg/mL, 0.1 µg/mL, and 1 µg/mL). Relative toxicity is calculated by normalizing CD1a x CD3-treated T-ALL cells co-cultured with PBMCs on negative control (co-cultures of T-ALL cells + PBMCs exposed to vehicle) placed equally to zero.

**Figure 3 cancers-14-02886-f003:**
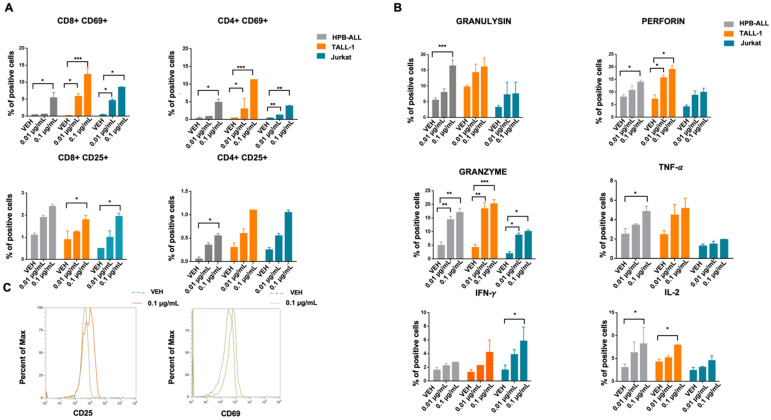
CD1a x CD3ε BTCE in vitro functional activity. (**A**) Activation markers (CD69, CD25) on CD8-CD4 T lymphocytes of vehicle or co-cultured with T-ALL cells at a 10:1 E:T ratio in the presence of vehicle or increasing concentration of CD1a x CD3ε BTCE. (**B**) Cytotoxic enzyme production (perforin, granzyme, and granulysin), cytokine release (IFN-y, TNF-a and IL-2) on T lymphocytes co-cultured with T-ALL cells in the presence of vehicle, or increasing concentration of CD1a x CD3ε BTCE. (**C**) Representative FACS data of CD25 and CD69 on cell membrane of PBMCs co-cultured with T-ALL cells at a 10:1 E:T ratio in the presence of vehicle or increasing concentration of CD1a x CD3ε BTCE; * *p* < 0.05, ** *p* < 0.01, *** *p* < 0.001.

**Figure 4 cancers-14-02886-f004:**
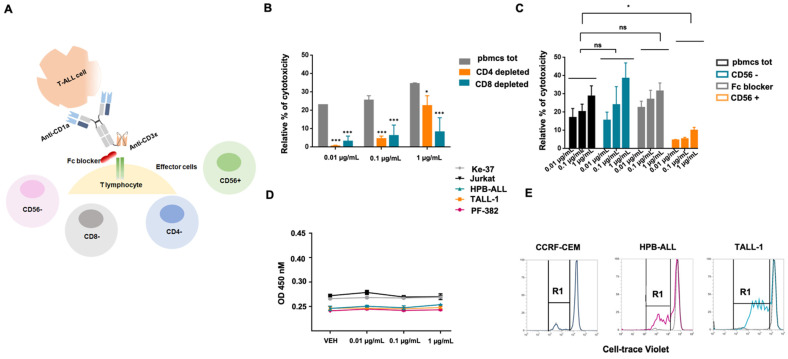
CD1a x CD3ε BTCE in vitro activity is mediated by T cell. (**A**) Schematic representation of experimental design. (**B**) Relative percentage (%) of T-cell redirected cytotoxicity of PBMCs, CD4, and CD8 depleted lymphocytes co-cultured with T-ALL cells in the presence of increasing concentrations of the BTCE. (**C**) Relative percentage (%) of T-cell redirected cytotoxicity of PBMCs, CD56 depleted, CD56 enriched, and PBMCS Fc blocked co-cultured with T-ALL cells in the presence of increasing concentration (0.01,0.1 and 1 µg/mL) of the BTCE. (**D**) CD1a x CD3ε BTCE direct cytotoxicity in the absence of effector cells on T-ALL cell lines. (**E**) Proliferation of T-cells co-cultured with CCRF-CEM, HPB-ALL, and TALL-1 cells, respectively, in the presence of the BTCE. * *p* < 0.05, *** *p* < 0.001.

**Figure 5 cancers-14-02886-f005:**
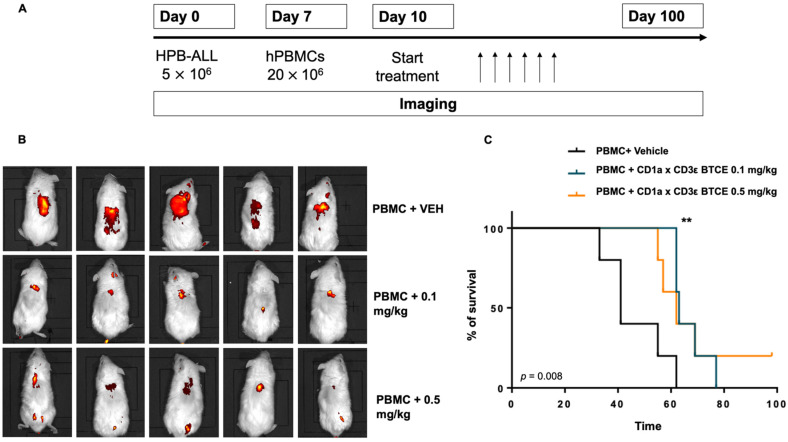
CD1a x CD3ε BTCE in vivo activity. (**A**) Schematic representation of in vivo experiment. (**B**) Fluorescence in vivo imaging of mice engrafted with human healthy donor derived PBMCs and subsequently treated with the BTCE at 0.1 or 0.5 mg/kg, as compared to vehicle at day 28. (**C**) Survival curves (Kaplan–Meier) of mice engrafted with human healthy donor derived PBMCs and treated with CD1a x CD3ε BTCE or vehicle (log-rank test, *p* < 0.05). ** *p* < 0.01.

**Figure 6 cancers-14-02886-f006:**
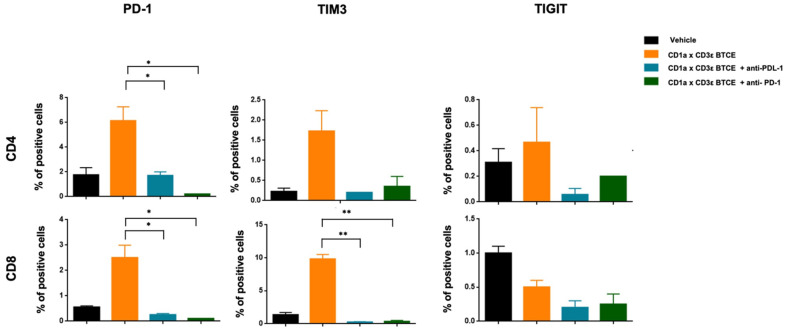
Immune checkpoint inhibitors and CD1a x CD3ε combinatorial approach. Expression of exhaustion markers (PD-1, TIM-3, LAG-3, and TIGIT) on CD4+ and CD8+ T-lymphocytes co-cultured with Jurkat cell line in the presence of CD1a x CD3 alone, CD1a x CD3ε plus anti-PD-1, or anti PD-L1. * *p* < 0.05, ** *p* < 0.01.

## Data Availability

RNA expression data were obtained from GSE69239 data set.

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
