# Peer review of "A Novel Bispecific T-Cell Engager (CD1a x CD3ε) BTCE Is Effective against Cortical-Derived T Cell Acute Lymphoblastic Leukemia (T-ALL) Cells"

_cancers, 2022, doi:10.3390/cancers14122886_

Round 1

Reviewer 1 Report

The authors in the manuscript entitled “A novel bispecific T-cell engager (CD1a x CD3e) BTCE is effective against cortical-derived T cell acute lymphoblastic leukemia (T-ALL) cells” describe the development and assessment of an anti-CD1axCD3 bispecific antibody to target and kill CD1a+ T ALL cells. Through various in vitro cell based assays, the authors show that the bispecific can induce activation, proliferation and killing of CD1a+ T ALL cells when cultured with PBMCs.  In an immunocompromised mouse model, the authors demonstrate that the bispecific can control growth of implanted T ALL tumor and improve the survival. The study is well executed and written. There are many CD3 directed bispecific modalities that are being tested in the clinic but in the setting of T ALL, a limited number of studies have been published. This study described in this manuscript would add value in furthering our understanding to tackle this disease. A few comments can be found below.

It would be good to show some biophysical characterization data of the anti-CD1axCD3 bispecific. There might be some readers who would like to see the data on the characterization of this bispecific. Perhaps the data can be included in section 3.1  where the title says “Generation and characterization of humanized UMG2 mAb”

Have you looked at the activity of this bispecific antibody towards primary T cells in the presence of CD1a expressing T ALL cells from patients? Is the cytotoxicity selective to T ALL but not T cells? In Fig1F, it seems clear that CD1a is not expressed on T cells so you would not expect cytotoxicity, but do you have data from an experiment done like Fig 4D but instead of using T ALL cells, using primary T cells and looking at the cytotoxicity in the presence of the bispecific?

Fig 2C the Kd value has not been reported either in the legend or on the figure even though the legend says “C) Average apparent CD1a x CD3. BTCE Kd evaluated on CD1a cortical derived T-ALL cell lines.”

Author Response

The authors in the manuscript entitled “A novel bispecific T-cell engager (CD1a x CD3e) BTCE is effective against cortical-derived T cell acute lymphoblastic leukemia (T-ALL) cells” describe the development and assessment of an anti-CD1axCD3 bispecific antibody to target and kill CD1a+ T ALL cells. Through various in vitro cell based assays, the authors show that the bispecific can induce activation, proliferation and killing of CD1a+ T ALL cells when cultured with PBMCs.  In an immunocompromised mouse model, the authors demonstrate that the bispecific can control growth of implanted T ALL tumor and improve the survival. The study is well executed and written. There are many CD3 directed bispecific modalities that are being tested in the clinic but in the setting of T ALL, a limited number of studies have been published. This study described in this manuscript would add value in furthering our understanding to tackle this disease. A few comments can be found below.

It would be good to show some biophysical characterization data of the anti-CD1axCD3 bispecific. There might be some readers who would like to see the data on the characterization of this bispecific. Perhaps the data can be included in section 3.1 where the title says “Generation and characterization of humanized UMG2 mAb”

We thank the Reviewer 1 for the careful evaluation of our manuscript and this important suggestion. Accordingly, in the present version of the manuscript, we have included HPLC characterization data of final anti-CD1axCD3 product that has been used for in vitro and in vivo study. These data are now included in the Section 2.7 of Material and Methods section lines 182-185.

Have you looked at the activity of this bispecific antibody towards primary T cells in the presence of CD1a expressing T ALL cells from patients? Is the cytotoxicity selective to T ALL but not T cells? In Fig1F, it seems clear that CD1a is not expressed on T cells so you would not expect cytotoxicity, but do you have data from an experiment done like Fig 4D but instead of using T ALL cells, using primary T cells and looking at the cytotoxicity in the presence of the bispecific?

We thank again the Reviewer for this further important point. In the light of a potential translation path, we believe it is necessary to clarify that no T cell fratricide activity has been observed even with patient primary T cells. Based on our available data, a new figure clarifying this point has been added in Supplementary data, showing that gated normal primary T cells are not affected by anti-CD1axCD3 treatment (Supplementary Fig.7A-B).

Fig 2C the Kd value has not been reported either in the legend or on the figure even though the legend says “C) Average apparent CD1a x CD3. BTCE Kd evaluated on CD1a cortical derived T-ALL cell lines.”

We apologize for this lack of information. In the present version of the manuscript, the Kd value has been reported in the legend as suggested.

Reviewer 2 Report

T-cell acute lymphoblastic leukemia (T-ALL) is an aggressive malignant neoplasm of the bone marrow. T-ALL accounts for only 10% to 15% of pediatric and up to 25% of adult ALL cases.Its clinical presentation can include hyperleukocytosis with extramedullary involvement of lymph nodes and other organs, including frequent central nervous system infiltration and the presence of a mediastinal mass, arising from the thymus. The WHO defines lymphoblasts in T-ALL as TdT positive with variable expression of CD1a, CD2, CD3, CD4, CD5, CD7, and CD8.  T-ALL can be subdivided into different stages by intrathymic differentiation, including pro-T, pre-T, cortical T, and medullary T. CD1a is expressed in only cortical T ALL and these T-ALL cases with TLX1 or NKX2.1 aberrations have been associated with excellent treatment outcomes. OS rates for adult patients with T-ALL are lower than 50% due to higher treatment-related toxicities. Patients are assigned to standard-, medium-, or high-risk group based on initial steroid response and minimal residual disease (MRD) after the first two courses of chemotherapy. Risk-based intensification of the therapeutic regimen has greatly improved the survival rate for pediatric and young adult patients treated on pediatric-based protocols. Nevertheless, still 1 of 5 pediatric patients with T-ALL dies within 5 years after first diagnosis from relapsed disease and therapy resistance (refractory disease) or from treatment-related mortalities, including toxicity and infections. Therefore, there is an urgent need for implementation of targeted therapies in high risk patients.  BH3 mimetics, NOTCH1 inhibitors, BET inhibitors, ABL/Src-Family Kinase Inhibitors, JAK Inhibitors, PIM1 inhibitor, PI3K-AKT-mTOR inhibitors, MEK Inhibitors, CDK Inhibitors are under investigation.  At the moment, nelarabine is the only novel drug approved for the treatment of relapsed T-ALL cases. As a single agent for relapsed or refractory T-ALL in children and young adults, nelarabine had a response rate of over 50%. In adults, these response rates were somewhat lower (36% achieved complete remission), but they still provided encouraging results for relapsed cases by inducing clinical remissions that facilitated access to stem cell transplantation. Genetically engineered autologous chimeric antigen receptor T (CAR T) cells have been used successfully as therapy for various malignancies including relapsed ALL. Initially, the challenge to harvest sufficient mature T cells from patients with T-cell malignancies without any lymphoblast contamination hampered the development of CAR T cells against T-ALL.  In addition, the fratricide effect—the paradigm that CAR T cells share the same surface markers with their malignant T-cell targets—would rapidly self-extinguish the CAR T cells. After the first approval of the anti-CD19 CAR T for the treatment of pediatric patients with relapsed B-ALL, many different surface proteins have been investigated for the development of novel CAR T therapies directed toward T-cell malignancies, including CD5, CD7, CD1, and CD38. CD1a is promising target for refractory or relapsed cortical T-ALL . Moreover, CD1a is expressed only during the proliferative phase of thymocyte development and not on immature progenitor cells or mature T cells, limiting the risk of complete immunodeficiency after treatment. Recently, the development of fratricide-resistant anti-CD1a CAR T cells for the treatment of CD1a-positive T-ALL has been reported.  This study shows a novel CD1a and CD3 BTCE (UMG2mAb) for the treatment of CD1a expressing T-ALL patients. There are several concerns about this manuscript are listed as follows:

  1. This study is very novel and crucial. However, because patients with CD1-positive cortical T-ALL have been associated with excellent outcomes, it is not known what percentage of patients with relapsed T-ALL will express CD1 and thus benefit from such BTCE therapy. Please discuss this in the discussion. 
  2. In figure 1C, please include the title of the Y axis instead of an abbreviation. 
  3. In figure 1D, please indicate at what ul of UMG2 is demonstrated in competition assay.  Instead of at the top of the plots, give the anti-CD1a clones and Anti-CD1aNA at the bottom. 
  4. In figure 2 and 3, please indicate the comparison of entities with brackets, it is hard to follow which statistically significant star compares what?
  5. In figure 2H what do the different colored columns represent?
  6. For in vivo studies, how did you decide on the animal model? TALL-1 in vitro results seems similar, why HPB-ALL cells were used? How did you decide on doses and scheme of administration? (Please refer to previous literature if any?)
  7. The increased concentration of UMG2 was advantageous in vitro but not in vivo, please add the bioluminescence graphic to show the significant reduction of tumor burden. 
  8. Regarding to expression on langerhans cells, what kind of side effects should be considered- off tumor toxicity?

Author Response

T-cell acute lymphoblastic leukemia (T-ALL) is an aggressive malignant neoplasm of the bone marrow. T-ALL accounts for only 10% to 15% of pediatric and up to 25% of adult ALL cases. Its clinical presentation can include hyperleukocytosis with extramedullary involvement of lymph nodes and other organs, including frequent central nervous system infiltration and the presence of a mediastinal mass, arising from the thymus. The WHO defines lymphoblasts in T-ALL as TdT positive with variable expression of CD1a, CD2, CD3, CD4, CD5, CD7, and CD8.  T-ALL can be subdivided into different stages by intrathymic differentiation, including pro-T, pre-T, cortical T, and medullary T. CD1a is expressed in only cortical T ALL and these T-ALL cases with TLX1 or NKX2.1 aberrations have been associated with excellent treatment outcomes. OS rates for adult patients with T-ALL are lower than 50% due to higher treatment-related toxicities. Patients are assigned to standard-, medium-, or high-risk group based on initial steroid response and minimal residual disease (MRD) after the first two courses of chemotherapy. Risk-based intensification of the therapeutic regimen has greatly improved the survival rate for pediatric and young adult patients treated on pediatric-based protocols. Nevertheless, still 1 of 5 pediatric patients with T-ALL dies within 5 years after first diagnosis from relapsed disease and therapy resistance (refractory disease) or from treatment-related mortalities, including toxicity and infections. Therefore, there is an urgent need for implementation of targeted therapies in high risk patients.  BH3 mimetics, NOTCH1 inhibitors, BET inhibitors, ABL/Src-Family Kinase Inhibitors, JAK Inhibitors, PIM1 inhibitor, PI3K-AKT-mTOR inhibitors, MEK Inhibitors, CDK Inhibitors are under investigation.  At the moment, nelarabine is the only novel drug approved for the treatment of relapsed T-ALL cases. As a single agent for relapsed or refractory T-ALL in children and young adults, nelarabine had a response rate of over 50%. In adults, these response rates were somewhat lower (36% achieved complete remission), but they still provided encouraging results for relapsed cases by inducing clinical remissions that facilitated access to stem cell transplantation. Genetically engineered autologous chimeric antigen receptor T (CAR T) cells have been used successfully as therapy for various malignancies including relapsed ALL. Initially, the challenge to harvest sufficient mature T cells from patients with T-cell malignancies without any lymphoblast contamination hampered the development of CAR T cells against T-ALL.  In addition, the fratricide effect—the paradigm that CAR T cells share the same surface markers with their malignant T-cell targets—would rapidly self-extinguish the CAR T cells. After the first approval of the anti-CD19 CAR T for the treatment of pediatric patients with relapsed B-ALL, many different surface proteins have been investigated for the development of novel CAR T therapies directed toward T-cell malignancies, including CD5, CD7, CD1, and CD38. CD1a is promising target for refractory or relapsed cortical T-ALL . Moreover, CD1a is expressed only during the proliferative phase of thymocyte development and not on immature progenitor cells or mature T cells, limiting the risk of complete immunodeficiency after treatment. Recently, the development of fratricide-resistant anti-CD1a CAR T cells for the treatment of CD1a-positive T-ALL has been reported.  This study shows a novel CD1a and CD3 BTCE (UMG2mAb) for the treatment of CD1a expressing T-ALL patients. There are several concerns about this manuscript are listed as follows:

This study is very novel and crucial. However, because patients with CD1-positive cortical T-ALL have been associated with excellent outcomes, it is not known what percentage of patients with relapsed T-ALL will express CD1 and thus benefit from such BTCE therapy. Please discuss this in the discussion.

We thank the Reviewer 2 for very careful evaluation of our work. We agree on the relevance of this comment. In the present version of the manuscript, this point has been highlighted in the discussion section 551-555.

In figure 1C, please include the title of the Y axis instead of an abbreviation. 

Changes have been made as suggested.

In figure 1D, please indicate at what ul of UMG2 is demonstrated in competition assay.  Instead of at the top of the plots, give the anti-CD1a clones and Anti-CD1aNA at the bottom. 

Changes have been made as suggested. In the current version of manuscript, the amount of UMG2 has been indicated in the figure legend.

In figure 2 and 3, please indicate the comparison of entities with brackets, it is hard to follow which statistically significant star compares what?

In the present version of the manuscript, we clarify that P values are calculated by comparing PBMCs + T-ALL co-cultured cells exposed to vehicle and PBMCS + T-ALL co-cultured cells treated with increasing concentration of CD1a x CD3. Relative toxicity is calculated by normalizing treated CD1a x CD3 T-ALL cells co-cultured with PBMC on negative control (T-ALL cells co-cultured with PBMCs and treated with vehicle) placed equally to zero. All these points have been now clarified in the legend. Moreover, we indicate the comparison of entities with brackets in Fig.3.

In figure 2H what do the different colored columns represent?

We thank the Reviewer for the question, and we apologize for the lack of clarity. In the current version of the manuscript, we add the relative legend to Fig 2H. Different colored columns represent increasing concentration of CD1a x CD3 BTCE and all data are now reported in the legend.

For in vivo studies, how did you decide on the animal model? TALL-1 in vitro results seem similar, why HPB-ALL cells were used? How did you decide on doses and scheme of administration? (Please refer to previous literature if any?)

We thank the Reviewer for the comment that allow us to clarify. We select HPB-ALL as target cells based on lab extensive experience for in vivo model generation. A previous reference is quoted. On this basis, we decide the dose, route and scheme of administration of CD 1a x CD3.

The increased concentration of UMG2 was advantageous in vitro but not in vivo, please add the bioluminescence graphic to show the significant reduction of tumor burden. 

Unfortunately, to reduce the number of animals in the experiment, based on restricted Institutional regulations, we were not able to perform a formal dose-response experiment. Since the concentrations used were very close, we did not detect any significant change at higher concentration. Moreover, for technical problems during the experiment, we were not able to generate a bioluminescence graph. 

Regarding to expression on Langerhans cells, what kind of side effects should be considered- off tumor toxicity?

We thank the Reviewer for rising this point. Taking into account the abundance of Langerhans cells in the skin, we believe that some potential skin toxicity can be expected by targeting CD1a in a phase I clinical study.

Reviewer 3 Report

This is a well written manuscript with some grammatical errors and typos. However, I have a couple major and some minor concerns that need to be addressed.

Major

Fig. 5 – PBMC alone control group is missing. These cells alone could generate an allogeneic response without the need for BiTEs. This needs to be rectified.

Fig. 6 is missing.

Minor

Line 292- replace ‘clinical’ with ‘clinically’

Line 350 – replace ‘contest’ with ‘context’

The Figures should have the same font in all panels and should be legible.

Fig. 2F – how did you distinguish between the 2 cell types in co-culture. Gating strategy to be provided

Fig. 2G – P value of which 2 conditions compared? What do you mean by relative toxicity? Relative to what? This should be indicated for all figures.

Do T-ALL cells express CD107, CD69 and CD25?

Author Response

This is a well written manuscript with some grammatical errors and typos. However, I have a couple major and some minor concerns that need to be addressed.

Major

Fig. 5 – PBMC alone control group is missing. These cells alone could generate an allogeneic response without the need for BiTEs. This needs to be rectified.

 We thank the Reviewer 2 for raising this point that, we agree, needs to be clarified. Indeed, the control group includes PBMC. Accordingly, to avoid misunderstanding in the interpretation of Fig. 5, in the present version of the manuscript, we have clarified this point in the legend to Fig. 5.

Fig. 6 is missing.

 We apologize for the lack of the figure; it has been probably due to a mistake in the file attachment process. The present version of the manuscript is complete.

Minor

Line 292- replace ‘clinical’ with ‘clinically’

Changes have been made as suggested.

Line 350 – replace ‘contest’ with ‘context’

 Changes have been made as suggested.

The Figures should have the same font in all panels and should be legible.

We thank the Reviewer for this important suggestion, in the current version of the manuscript we have uniformed all font in all figure panels.

Fig. 2F – how did you distinguish between the 2 cell types in co-culture. Gating strategy to be provided.

We agree also on this point. In the present version of the manuscript, the gating strategy to distinguish between the two cell types in co-culture has been detailed in 2.9 Material and Methods section.

Fig. 2G – P value of which 2 conditions compared? What do you mean by relative toxicity? Relative to what? This should be indicated for all figures.

We thank the reviewer for this important suggestion. To improve the readability of the figures, in the present version of the manuscript we clarify that P value is calculated by comparing PBMCs + T-ALL cells exposed to vehicle and PBMCS + T-ALL cells treated with increasing concentration of CD1a x CD3e. Relative toxicity is calculated by normalizing treated CD1a x CD3e T-ALL cells co-cultured with PBMC on negative control (T-ALL cells co-cultured with PBMCs and treated with vehicle).

Do T-ALL cells express CD107, CD69 and CD25?

We thank the Reviewer for careful evaluation of our manuscript. Consistently with their ontology, T-ALL cells express these markers.

Round 2

Reviewer 3 Report

The authors have addressed majority of the concerns. However, these other concerns based on the fig 6 which is now provided, need to be addressed.

The authors have indicated in their response that Consistently with their ontology, T-ALL cells express these markers. If both T cells and T-ALL cells express CD69, CD25 and CD107, how are the authors able to differentiate between the two cell types?

Fig. 5 legend revise to state that the mice were engrafted with both T-ALL and T cells. Why was "untreated" group not included? These would be the mice only engrafted with T-ALL cells.

PD-L1 and PD-1 inhibitor labels are not consistent between Fig. 6 and supp fig 9. Legend to supp fig 9 is missing. Was the expression of PD-L1 on CD1a expressing cells evaluated post treatment? why is the percentage of PD-L1 cells higher in nivolumab treated cells? I feel it is premature to make the conclusion just based on in vitro data.

line 456 - insert 'in' between reduction and expression.

line 459 - change 'increase' to 'increased'

Round 3

Reviewer 3 Report

All concerns addressed, thank the authors for revising the manuscript.